# Geographical Distribution of *Borrelia burgdorferi* sensu lato in Ticks Collected from Wild Rodents in the Republic of Korea

**DOI:** 10.3390/pathogens9110866

**Published:** 2020-10-22

**Authors:** Seong Yoon Kim, Tae-Kyu Kim, Tae Yun Kim, Hee Il Lee

**Affiliations:** Division of Vectors and Parasitic Diseases, Bureau of Infectious Disease Diagnosis Control, Korea Disease Control and Prevention Agency, 187 Osongsaengmyeong 2-ro, Cheongwon-gun, Cheongju-si 363-951, Chungcheongbuk-do 28159, Korea; gunbo0402@korea.kr (S.Y.K.); tkkim80@korea.kr (T.-K.K.); kty4588@korea.kr (T.Y.K.)

**Keywords:** lyme disease, *Borrelia burgdorferi* sensu lato, tick, wild rodent

## Abstract

Lyme disease is a tick-borne zoonotic disease caused by *Borrelia burgdorferi* sensu lato (s. l.) via transmission cycles involving competent tick vectors and vertebrate reservoirs. Here, we determined the prevalence and distribution of *Borrelia* genospecies in 738 ticks of at least three species from wild rodents in nine regions of the Republic of Korea (ROK). Ticks were analyzed using nested PCR targeting partial flagellin B gene sequences, followed by sequence analysis. The prevalence of *Borrelia* infection was 33.6%, and the most common genospecies were *B. afzelii* (62.5%), *B. valaisiana* (31.9%), *B. yangtzensis* (2.4%), *B. garinii* (1.6%), and *B. tanukii* (1.6%). *Borrelia afzelii* was found in all regions except Jeju Island; this predominant genospecies was found in the northern and central sampling regions. *Borrelia valaisiana*, *B. yangtzensis*, and *B. tanukii* were found only in the southern regions with *B. valaisiana* being the most common, whereas *B. yangtzensis* and *B. tanukii* were only found on Jeju Island. Our study is the first to describe the nationwide prevalence of *B. burgdorferi* s. l. in ticks from wild rodents in the ROK. Continuous surveillance in ticks, animals, humans, and different regions is required to avoid disease distribution and possible transmission to humans in the ROK.

## 1. Introduction

Lyme disease (LD) is the most prevalent tick-borne zoonotic disease and is common in all temperate regions of the Northern Hemisphere, including Asia, Europe, America, Africa, and Oceania, with more than 0.3 million clinical cases per year [1]. *Borrelia burgdorferi* sensu lato (s. l.) is the causative agent of LD and comprises of at least 20 different named and unnamed genospecies [2]. Three of these are the major causative agents of LD in humans and display different clinical manifestations and geographical distributions. For instance, *B. afzelii* and *B. garinii* are known to cause dermatitis and neuritis, respectively, in Eurasia, while *B. burgdorferi* sensu stricto (s. s) causes arthritis in the United States and Western Europe [3,4,5].

*Borrelia burgdorferi* sensu lato (s. l.) is mainly maintained via transmission cycles involving competent tick vectors and vertebrate reservoirs [6]; therefore, ticks are used for epidemiological studies of tick-borne pathogens [7]. *Ixodes ricinus* is typically found in Europe, while *I. scapularis* and *I. pacificus* are found in North America, and *I. persulcatus* and *I. nipponensis* are found in Asia [4]. The larvae of hard ticks are uninfected due to the lack of transovarial transmission; however, *B. burgdorferi* s. l. is acquired by feeding on an infected animal reservoir. After molting to the next stage, the ticks can transmit the pathogen to animals such as small mammals, which are common hosts for immature ticks [2]. Since blood-sucking vectors contain infected host blood and the pathogen itself, they are reliable tools for demonstrating the existence of the pathogens in a specific area [7]. Moreover, micromammal species have been widely investigated because they act as natural reservoirs in the *Borrelia* transmission cycle [8].

*Borrelia burgdorferi* sensu lato (s. l.) was first identified in the Republic of Korea (ROK) in *Ixodes* ticks and the wild rodent *Apodemus agrarius* in 1993, with the first human case of LD reported in the same year [9,10]. Lyme disease (LD) was designated as a national notifiable infectious disease in 2010, and two cases were identified in 2011. Since then, the number of cases has gradually increased; 23 cases have been reported in 2019, including 11 imported cases [11]. To date, various molecular and serological studies have identified the pathogenic cause of LD in ticks and animals including wild rodents, migratory birds, water deer, and dogs [12,13,14,15,16,17,18,19,20]. *Borrelia afzelii* and *B. garinii* were first identified in *I. persulcatus* and wild rodents, respectively, from the Gangwon province. Moreover, other genospecies of *B. burgdorferi* s. l., including *B. tanukii*, *B. turdi*, *B. yangtzensis*, *B. bavariensis*, and *B. valaisiana,* have been identified in ticks and wild animals [9,16,18,19,20,21,22]. Regional information regarding ticks, wild animals, and *B. burgdorferi* s. l. genospecies has been revealed sporadically in the ROK; however, the main *B. burgdorferi* s. l. genospecies and its prevalence within tick distributions remain unknown.

In this study, we aimed to determine the prevalence and distribution of *Borrelia* genospecies in ticks attached to wild rodents throughout ROK as part of a nationwide investigation.

## 2. Results

### 2.1. Tick Samples and Infection Rates

First, we examined *Borrelia* infection in engorged ticks attached to 123 wild rodents collected from nine regions. *Apodemus agrarius* (90.2%) was the predominant rodent species, followed by *Crocidura lasiura* (8.9%) and *Craseomys regulus* (0.8%; data not shown). A total of 738 engorged ticks were collected that included ticks from two genera and at least three species. Some *Ixodes* spp. cannot be identified in their larval stage so we used the term “*Ixodes* spp.” to refer to these ticks (Table 1). *Ixodes nipponensis*, *I. angustus*, and *H. longicornis* were collected from wild rodents and examined individually. Overall, 33.6% (248 positive ticks among the 738 ticks examined) of the ticks tested positive via PCR, including 56.5% of *I. nipponensis*, 30.7% of *Ixodes* spp., 11.1% of *H. longicornis*, and 3.1% of *I. angustus* (Table 1).

### 2.2. Molecular Identification of B. burgdorferi s. l.

A total of five *B. burgdorferi* s. l. genospecies were identified, of which the most common genospecies were *B. afzelii* (62.5%), followed by *B. valaisiana* (31.9%), *B. yangtzensis* (2.4%), *B. garinii* (1.6%), and *B. tanukii* (1.6%; Table 1). In the regional distribution, *B. afzelii* was found in all the inland sampling regions except for Jeju Island, while Sejong and Uiseong located in the central inland regions only identified *B. afzelii*. *Borrelia valaisiana* was found in Geoje, Goheung, and Jeju Island, which are located in the southern sampling regions, and displayed a higher prevalence than other genospecies. Specifically, *B. yangtzensis* and *B. tanukii* were only found on Jeju Island (Figure 1).

### 2.3. Phylogenetic Tree of B. burgdorferi s. l.

We selected 24 representative samples without duplicate sequences among the 248 ticks identified as positive for *Borrelia* genospecies and constructed a phylogenetic tree using the flagellin B (*flaB*) genes from several *Borrelia* genospecies sequences deposited in GenBank (Figure 2). The representative sequences of the five genospecies clustered with related reference sequences; however, *B. yangtzensis* clustered together in a large clade closely related to *B. valaisiana*. No apparent clustering was observed based on the tick host or sampling region.

## 3. Discussion

In this study, we performed the molecular detection and phylogenetic analysis of *B. burgdorferi* s. l. in engorged ticks from wild rodents in nine regions of the ROK. Nested PCR revealed an overall *Borrelia* infection rate of 33.6% among the 738 ticks analyzed, similar to the high prevalence rates found in ticks feeding on wild rodents in Malaysia (46.1%) and Taiwan (47.1%) [23,24]. However, previous studies from the ROK have detected very low rates of *B. burgdorferi* s. l. in ticks feeding on wild animals such as dogs (0.2%, *n* = 562 ticks) [20], wild water deer (2.1%, *n* = 48 ticks) [19], migratory birds (3.7%, four positive pools of 108 tested pools among 212 ticks) [18], and wild rodents in northern Gyeonggi near the demilitarized zone (1.0%, *n* = 1618 ticks including 933 questing ticks) [25]. These differences in prevalence may be due to variations in the sampling methods for tick and host species, survey regions, the environment, and the seasonal timing of the survey. 

Sequence analysis, including phylogenetic analysis based on partial *flaB* sequences, confirmed that *B. afzelii*, *B. valaisiana*, *B. garinii*, *B. yangtzensis*, and *B. tanukii*, were detected in the ticks collected in this study and displayed isolated geographical distribution. For instance, *B. afzelii* and *B. garinii* were found only inland, while *B. afzelii* was the main genospecies in the northern and central sampling regions. Conversely, *B. valaisiana*, *B. yangtzensis*, and *B. tanukii* were only found in the southern sampling regions, where *B. valaisiana* was the most common genospecies. Previous studies in the ROK have identified *B. afzelii* and *B. garinii* in various tick species, animals, and patient sera throughout inland areas [9,16,19,20,21,22,26]. In addition, other *B. burgdorferi* s. l. groups, such as *B. valaisiana*, have been detected in *I. nipponensis* [27], whereas *B. yangtzensis* was reclassified via multilocus sequence typing in *A. agrarius* from Haenam in the southernmost inland region [22]. *Borrelia tanukii* has been reported in *Ixodes tanuki* from Japan [28]. In the ROK, a similar partial 16s rRNA sequence of *B. tanukii* has been reported in ticks feeding on migratory birds on Hongdo Island [18]; however, it was unclear whether the ticks were indigenous. Our study demonstrates for the first time that *B. tanukii* is indigenous to ticks in the ROK. Many *B. burgdorferi* s. l. genospecies, including *B. afzelii*, *B. garinii*, *B. burgdorferi* s. s., *B. bavariensis*, and *B. bissettii,* are known to be pathogenic to humans, while *B. lusitaniae*, *B. spielmanii*, and *B. valaisiana* are considered potentially pathogenic [3]. In this study, the identification of *B. afzelii* and *B. valaisiana* in ticks that act as major *Borrelia* vectors indicates that LD transmission may be possible via questing ticks in the ROK.

A previous study in which vegetation was swept and dragged for questing ticks identified *H. longicornis* and *H. flava* as the dominant tick species, while *I. nipponensis* has been shown to be the most frequently collected tick from small mammals [29,30]. Our study found that most of the ticks collected from wild rodents were of the *Ixodes* genus, including *Ixodes* spp. (*n* = 440, 59.6%), *I. nipponensis* (*n* = 193, 26.1%), and *I. angustus* (*n* = 96, 13.0%). Moreover, we found that *Borrelia* DNA was prevalent in *I. nipponensis*, *Ixodes* spp., *I. angustus*, and *H. longicornis*; however, the prevalence varied between species. *Borrelia burgdorferi* sensu lato (s. l.) was first identified from *I. angustus* in the ROK and has since been found in several other tick species, such as *I. granulatus*, *I. persulcatus*, *I. nipponensis*, *I. turdus*, and *H. longicornis* [9,18,19,27]. In experimental vector competence studies, *I. angustus* was found to be a competent vector for transmitting *B. burgdorferi* s. s. to deer mice [31], while this study identified *Borrelia* infection in 1/9 *H. longicornis* larvae. A previous study reported the first molecular detection of *B. afzelii* in *H. longicornis* infesting wild Korean water deer (*Hydropotes inermis*) in the ROK [19]; however, Sun et al. [32] showed that *H. longicornis* can carry but not transmit *Borrelia* as *H. longicornis* can only maintain spirochetes for a shorter period than the digestion period of blood. Therefore, *H. longicornis* is not the main *Borrelia* vector, despite being dominant in the ROK, indicating that *Ixodes* species may be the main vector and carrier of *Borrelia* in the ROK. Because we used ticks harvested from wild rodents, the *Borrelia* genospecies and infection rate in this study may have originated from the wild rodent itself. Nevertheless, blood-sucking vectors are reliable tools for demonstrating the existence of pathogens in a specific area [7]. Human biting cases by the *Ixodes* species was significantly lower (5.7%) than those by other species in the ROK [33]. Considering the gradually increasing number of LD cases in the ROK and the high *Borrelia* infection rate in ticks engorged with wild rodent blood in this study, monitoring for *Borrelia* in non-engorged or questing ticks at different developmental stages should be carried out. 

To our knowledge, this study is the first to describe the nationwide prevalence of *B. burgdorferi* s. l. in engorged ticks from wild rodents in the ROK. The ticks were found to have a high infection rate (33.6%) and a wide geographical distribution for the five detected genospecies of *B. burgdorferi* s. l., the causative agent of LD. Humans working in agricultural fields, visiting mountains and reservoirs for recreational activities, or inhabiting residential areas may therefore be at risk of exposure to and infection with *Borrelia* due to close contact with wild rodents and ticks. Thus, continuous surveillance on various tick species, animals, humans, and different geographical regions is required to reduce disease distribution and possible transmission to humans in the ROK.

## 4. Materials and Methods

### 4.1. Surveillance Localities and Periods

Ticks were collected from wild rodents during spring (March–May) and autumn (September–November) in 2017 from nine regions in the ROK, namely Pocheon, Donghae, Sejong, Boryeong, Uiseong, Jeongup, Geoje, Goheung, and Jeju Island (Figure 1). Rodent traps were installed at several environmental sites, including cropped fields, reservoirs, waterways, and mountains. The animal-handling protocol used in this study was reviewed and approved based on the guidelines for ethical procedures and scientific care of the Institutional Animal Care and Use Committee of the Korea Centers for Disease Control and Prevention (KCDC-089-17).

### 4.2. Collection of Wild Rodents and Ticks

At each site, 25 Sherman folding live traps (3 × 3.5 × 9 inches, BioQuip, Gardena, CA, USA) were set up at five points at 3–5 m intervals with a peanut butter-spread biscuit. The rodent traps were collected the next morning and transported to the laboratory. After euthanization using compressed carbon dioxide, the rodents were suspended for 24 h over glass bowls filled with 70% EtOH to collect ticks. The ticks were then recovered from the surface using a fine brush and stored in 70% EtOH at −20 °C for identification.

### 4.3. Identification of Ticks

The species and developmental stage of each tick were identified under a dissection microscope using taxonomic identification keys according to Yamaguti et al. (1971) [34]. Individual ticks were then placed in a 2 mL screw cap tube according to the host rodent, collection site, region, and season and were stored at −20 °C before DNA extraction.

### 4.4. Detection of Borrelia DNA in Ticks

Tick samples were homogenized in 450 µL of 1× phosphate-buffered saline (PBS) with 2.8 mm ceramic (zirconium oxide) beads using a Precellys Evolution homogenizer (Bertin Technologies, Bretonneux, France). The homogenates were centrifuged at 25,000× *g* for 10 min, and 400 µL of the supernatant was used for DNA extraction using a MagMAX™ DNA Multi-Sample Ultra 2.0 Kit (Applied Biosystems, Foster city, CA, USA) according to the manufacturer’s instructions. To detect *Borrelia* DNA in tick samples, we targeted a partial *flaB* sequence, as described previously [34] (Table 2). For the first PCR reaction, 5 μL of tick template DNA was added to AccuPower^®^ HotStart PCR PreMix (Bioneer, Daejeon, Korea) and incubated at 94 °C for 5 min, followed by 35 cycles of 94 °C for 30 s, 60 °C for 45 s, and 72 °C for 1 min, with a final amplification at 72 °C for 10 min. Secondary nested PCR was then performed using 1 µL of the primary PCR products under the following conditions: 94 °C for 5 min, followed by 25 cycles of 94 °C for 10 s, 48 °C for 1 min, 72 °C for 90 s, and a final extension step at 72 °C for 5 min. Positive and negative controls were included in each PCR set. Amplified PCR products were visualized via 2.0% agarose gel electrophoresis after staining with Safe-Pinky DNA Gel Staining Solution (10,000×) in water (GenDEPOT, Barker, TX, USA), yielding a predicted size of 347 bp. To avoid cross-contamination, DNA extraction, amplification, and agarose gel electrophoresis were performed in separate rooms.

### 4.5. Sequencing and Phylogenetic Analysis

All positive nested PCR products were sequenced in directions, using the PCR primers at Bioneer Inc. (Daejeon, ROK). To identify *Borrelia* genospecies, the sequencing results were analyzed using the BlastN program from the National Center for Biotechnology Information (NCBI, Bethesda, MD, USA). The sequences obtained in this study were aligned using CLUSTALW and compared to published sequences in GenBank (NIH, Montgomery, MD, USA). A phylogenetic tree was constructed based on the DNA sequences of *flaB* from tick samples with a *Borrelia* infection using the neighbor-joining method with the p-distance model in MEGA version 5.2. Bootstrap analysis was conducted using 1000 replicates to improve the confidence level of the phylogenetic tree. The GenBank accession numbers of the genospecies sequences obtained in this study are presented in Figure 2.

## Figures and Tables

**Figure 1 pathogens-09-00866-f001:**
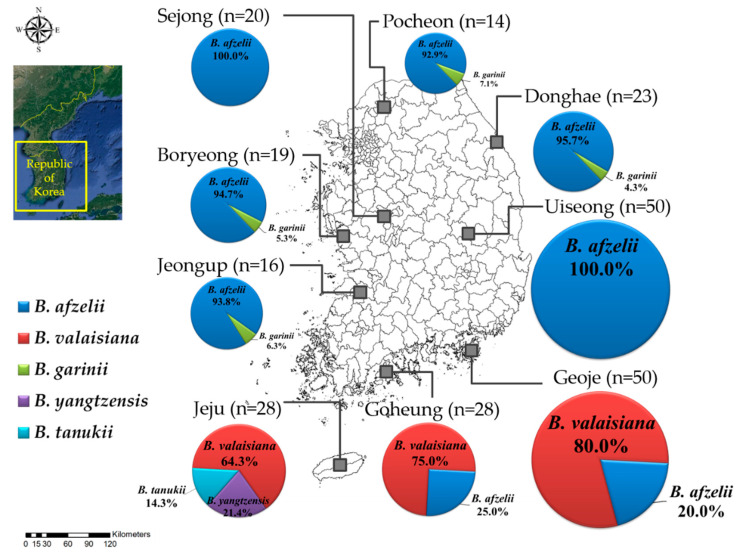
A map of the *Borrelia* genospecies detected in the different sampling regions of the ROK.

**Figure 2 pathogens-09-00866-f002:**
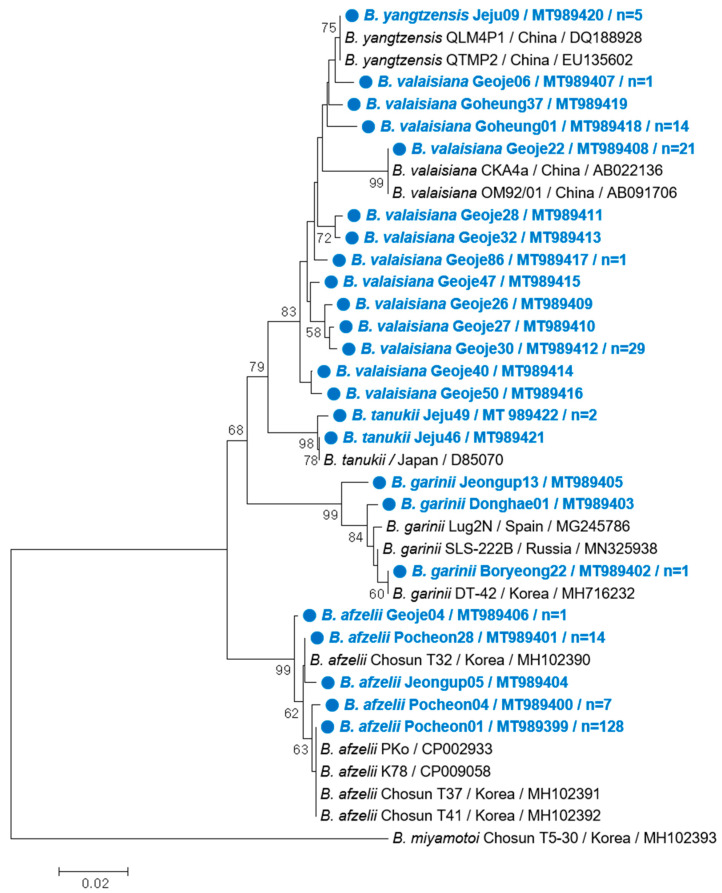
Phylogenetic relationship between *Borrelia* genospecies based on partial *flaB* nucleotide sequences (303 nucleotide positions) constructed using the neighbor-joining method in MEGA 5.2. Node numbers indicate the proportion of bootstrap replicates (1000) that supported the topology shown. The cutoff value for the consensus tree was 50%. The scale bar represents 2% divergence. The sequences identified in this study are indicated by blue circles. The number of sequences (*n*) with an identical genospecies are shown if the sequence was detected in more than one case.

**Table 1 pathogens-09-00866-t001:** Prevalence of *Borrelia* genospecies in engorged ticks via nested PCR for *flaB.*

Species	Stage	No. of Ticks	Positive (%)		*Borrelia* Genospecies				
					*B. afzelii*	*B. garinii*	*B. valaisiana*	*B. yangtzensis*	*B. tanukii*
*Ixodes* spp.	larva	440	135	(30.7)	102	1	29	3	0
*I. nipponensis*	nymph	193	109	(56.5)	53	3	50	2	1
*I. angustus*	larva	78	2	(2.6)	0	0	0	0	2
	nymph	18	1	(5.6)	0	0	0	0	1
*Haemaphysalis* *longicornis*	larva	9	1	(11.1)	0	0	0	1	0
Total		738	248	(33.6)	155	4	79	6	4

**Table 2 pathogens-09-00866-t002:** DNA sequences of primers used for nested PCR.

Targe Specificity	Designation	Sequence		Amplicon Length (bp)	Reference
**Flagellin B**	*BflaPAD*	1st PCR	5′-GATCARGCWCAAYATAACCAWATGCA-3′	459	[35]
	*BflaPDU*		5′-AGATTCAAGTCTGTTTTGGAAAGC-3′		
	*BflaPBU*	2nd PCR	5′-GCTGAAGAGCTTGGAATGCAACC-3′	347	
	*BflaPCR*		5′-TGATCAGTTATCATTCTAATAGCA-3′		

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
