# Peer review of "Geographical Distribution of Borrelia burgdorferi sensu lato in Ticks Collected from Wild Rodents in the Republic of Korea"

_pathogens, 2020, doi:10.3390/pathogens9110866_

Round 1

Reviewer 1 Report

The study is well described and structured and easy to followed.

Few comments from my side

Line 67: I would write: “were positive in the PCR”

Is not clear why the number 527 in Table 1 is not similar to Table 2. Maybe the authors can explain this in more detail for the readers of the journal?

Line 73-78: Maybe the authors can discuss these results in more detail for the readers of the journal.

Because the aim of the study is to determine the prevalence, I would recommend to include the term “prevalence” also in the Table 1 and 2.

Author Response

Response to reviewer’s comments

Manuscript ID: pathogens-956089

Title: Geographical distribution of Borrelia burgdorferi sensu lato in ticks collected from wild rodents in the Republic of Korea

Comments and Suggestions for Authors,

The study is well described and structured and easy to followed.

Few comments from my side.

Reply to the editor’s comments: Thank you for providing us this opportunity to further revise this manuscript. We appreciate the very positive and constructive comments and believe that we had addressed all the concerns/questions raised by the reviewer. We tried to do our best in the preparation of this revised manuscript.

Response to Reviewer 1 Comments

Point 1: Line 67: I would write: “were positive in the PCR”

Response 1: Following your comments and suggestion, we revised the phrase in Line 69-70.

Point 2: Is not clear why the number 527 in Table 1 is not similar to Table 2. Maybe the authors can explain this in more detail for the readers of the journal?

Response 2: The number 527 in Table 1 represents only the number of tested larva stage ticks. The number 738 in Table 2 represents total number of tested ticks. And moreover, we merged the table 1 and 2 according to the other reviewer’s comment.

Point 3: Line 73-78: Maybe the authors can discuss these results in more detail for the readers of the journal.

Response 3: We described the distribution of Borrelia genospecies by region briefly in Line 79-81. And we discussed previously reported distribution of Borrelia genospecies in ROK, and emphasized the first identification of Borrelia genospecies in ticks on rodents of ROK.

Point 4: Because the aim of the study is to determine the prevalence, I would recommend to include the term “prevalence” also in the Table 1 and 2.

Response 4: We included the term “prevalence” in the merged Table 1.

Reviewer 2 Report

The paper of Kim et al. presents the results of extensive study on Borrelia burgdorferi s.l. prevalence in various regions of South Korea Republic. When compared with previous studies, it shows significantly higher rate of Borrelia burgdorferi s.l.- infected ticks in most regions studied. It documents presence of rare Borrelia species, which have not yet been identified in the region, and different Borrelia species distribution when compared northern, central and southern regions.

The study brings interesting and epidemiologically important data that are worth publishing and fulfil the scope of Pathogens journal. The results are comprehensively written and supplemented with appropriate Figures and Tables. The methods used were appropriate for the purpose and mostly well described. The paper contains adequate discussion of the results and important literature data  the concerning the scope are cited both in the Introduction and Discussion sections.

Main comments:

Results: Statement that Ixodes sp. only was found in larvae stadium does not correspond to the Table 1 ( 9 larvae of Haemophysalis , one of them being infected).

Discussion: Haemophysalis is declared as a dominant tick species in Korea (line 142) but the results obtained in the study do not support this statement.

Authors should discuss disproportion between high circulation of Borreliae in environment and low incidence of Lyme borreliosis in Korea, as mentioned in the Introduction ( line 49,50).

Methods: Sequencing and phylogenetic analysis  deserve more detailed description.

Minor comments:

 Figure 2: The sequences coming from the study should be graphically distinguished from the  comparative sequences.

Repeated sentence in the Discussion  ( lines 108-9 and 149-50) should be omitted.

Methods: size of the PCR product in the text (line 193) differ from the size in the Table 2

Author Response

Response to reviewer’s comments

Manuscript ID: pathogens-956089

Title: Geographical distribution of Borrelia burgdorferi sensu lato in ticks collected from wild rodents in the Republic of Korea

Comments and Suggestions for Authors,

The paper of Kim et al. presents the results of extensive study on Borrelia burgdorferi s.l. prevalence in various regions of South Korea Republic. When compared with previous studies, it shows significantly higher rate of Borrelia burgdorferi s. l.- infected ticks in most regions studied. It documents presence of rare Borrelia species, which have not yet been identified in the region, and different Borrelia species distribution when compared northern, central and southern regions.

The study brings interesting and epidemiologically important data that are worth publishing and fulfil the scope of Pathogens journal. The results are comprehensively written and supplemented with appropriate Figures and Tables. The methods used were appropriate for the purpose and mostly well described. The paper contains adequate discussion of the results and important literature data the concerning the scope are cited both in the Introduction and Discussion sections.

Reply to the editor’s comments: Thank you for providing us this opportunity to further revise this manuscript. We appreciate the very positive and constructive comments and believe that we had addressed all the concerns/questions raised by the reviewer. We tried to do our best in the preparation of this revised manuscript.

Response to Reviewer 2 Comments

Point 1: Results: Statement that Ixodes sp. only was found in larvae stadium does not correspond to the Table 1 (9 larvae of Haemophysalis, one of them being infected).

Response 1: In this study, Ixodes spp. represents some larva stage unidentifiable ticks belonging to the Genus Ixodes taxonomically. Some Ixodes spp. in Far East can’t be identified in their larval stages (Yamaguti et al., 1971). So, we used the term Ixodes spp. to the unidentifiable larva of genus Ixodes in this manuscript. However, we can identified larval stage of Ixodes angustus and Haemaphysalis longicornis. We have inserted an additional explanation in the Line 67-8.

Point 2: Discussion: Haemophysalis is declared as a dominant tick species in Korea (line 142) but the results obtained in the study do not support this statement.

Response 2: As described in part of Discussion (Line 134-36), H. longicornis was dominant tick species in collecting host-seeking ticks by flagging and CO2 baited trap. However I. nippnensis including to the Genus Ixodes has been shown to the most frequently collected tick from small mammals in ROK (reference 29-30 in this manuscript). Ticks in this study were only collected from wild rodents.

Our data showed that most of the tick collected in wild rodents were genus Ixodes species including Ixodes spp. (n = 440, 59.6%), I. nipponensis (n = 193, 26.1%), and I. angustus (n = 96, 13.0%). Therefore, the results of this study are similar compared to previous study in ROK.

Point 3: Authors should discuss disproportion between high circulation of Borreliae in environment and low incidence of Lyme borreliosis in Korea, as mentioned in the Introduction (line 49, 50).

Response 3: We described the reason of disproportion between high circulation of Borreliae in environment and low incidence of Lyme borreliosis in Korea briefly in Line 153-8.

Point 4: Methods: Sequencing and phylogenetic analysis deserve more detailed description.

Response 4: We added more detailed description in Sequencing and phylogenetic analysis (line 207-14).

Point 5: Figure 2: The sequences coming from the study should be graphically distinguished from the comparative sequences.

Response 5: We revised the colors of the 24 representative sequence in the Figure 2.

Point 6: Repeated sentence in the Discussion (lines 108-9 and 149-50) should be omitted.

Response 6: We omitted line 114-116 of the overlapping sentences.

Point 7: Methods: size of the PCR product in the text (line 193) differ from the size in the Table 2

Response 7: We revised the 347 bp PCR product size in the Table 2.

Reviewer 3 Report

The manuscript by Kim et al. tilted“Geographical Distribution of Borrelia Burgdorferi Sensu Lato in Ticks Collected from Wild Rodents in the Republic of Korea” described the molecular detection of B. burgdorferi s. l. in ticks from wild rodents sampled across ROK. This study helped us to have a glimpse of the  nationwide prevalence of Borrelia Burgdorferi Sensu Lato in ticks from ROK.

Comments:

”Abbreviation should not be used at the beginning of a sentence, as in line 46 and throughout the manuscript.

Line 59, there is a omit of letter “o” (attach to).

Line 65 as well as Table 1, what does Ixodes spp. mean? Do the authors mean tick species unidentified?

Line 66, the word “and” should not be italic.

Table 1 is not easy to read. Throughout the manuscript the authors did not give any information about the infection rate of Borrelia in rodents where the ticks of this study were collected. So I don’t think it is necessary to arrange the table based on rodents species.

In line 67, “Overall, 33.6% (248) of the ticks were PCR positive...”, in the parentheses, it’s better to give the positive number as well as the total sample number.

Table 2 and Figure 1 was the same thing. So I suggest the authors to integrate Table 1 and 2 according to tick species. In Figure 1, it’s better to draw the pie chart according to the sample sizes in each sampling area. Besides, the number of ticks tested in Table 1 was 527, but in Table 2 was 738, please verify this.

Line 84, the manuscript was written in a passive form, this sentence was active.

Line112-115, the distribution of Borrelia might also be a result of the small sample size.

Author Response

Response to reviewer’s comments

Manuscript ID: pathogens-956089

Title: Geographical distribution of Borrelia burgdorferi sensu lato in ticks collected from wild rodents in the Republic of Korea

Comments and Suggestions for Authors,

The manuscript by Kim et al. tilted “Geographical Distribution of Borrelia Burgdorferi Sensu Lato in Ticks Collected from Wild Rodents in the Republic of Korea” described the molecular detection of B. burgdorferi s. l. in ticks from wild rodents sampled across ROK. This study helped us to have a glimpse of the nationwide prevalence of Borrelia Burgdorferi Sensu Lato in ticks from ROK.

Reply to the editor’s comments: Thank you for providing us this opportunity to further revise this manuscript. We appreciate the very positive and constructive comments and believe that we had addressed all the concerns/questions raised by the reviewer. We tried to do our best in the preparation of this revised manuscript.

Response to Reviewer 3 Comments

Point 1: Abbreviation should not be used at the beginning of a sentence, as in line 46 and throughout the manuscript.

Response 1: We revised the abbreviation in Line 36, 47, 49, 53, 68, 81, 126, and 141.

Point 2: Line 59, there is an omit of letter “o” (attach to).

Response 2: We revised the letter in Line 60.

Point 3: Line 65 as well as Table 1, what does Ixodes spp. mean? Do the authors mean tick species unidentified?

Response 3: In this study, Ixodes spp. represents some larva stage unidentifiable ticks belonging to the Genus Ixodes taxonomically. Some Ixodes spp. in Far East can’t be identified in their larval stages (Yamaguti et al., 1971). So, we used the term Ixodes spp. to the unidentifiable larva of genus Ixodes in this manuscript. However, we can identified larval stage of Ixodes angustus and Haemaphysalis longicornis. We have inserted an additional explanation in the Line 67-8.

Point 4: Line 66, the word “and” should not be italic.

Response 4: We revised the word in Line 68.

Point 5: Table 1 is not easy to read. Throughout the manuscript the authors did not give any information about the infection rate of Borrelia in rodents where the ticks of this study were collected. So I don’t think it is necessary to arrange the table based on rodents species.

Response 5: Following point 7 of your comments, we merged Table 1 and 2.

Point 6: In line 67, “Overall, 33.6% (248) of the ticks were PCR positive...”, in the parentheses, it’s better to give the positive number as well as the total sample number.

Response 6: We revised the parentheses in the Line 69-70.

Point 7: Table 2 and Figure 1 was the same thing. So I suggest the authors to integrate Table 1 and 2 according to tick species. In Figure 1, it’s better to draw the pie chart according to the sample sizes in each sampling area. Besides, the number of ticks tested in Table 1 was 527, but in Table 2 was 738, please verify this.

Response 7: The number 527 in Table 1 represents only the number of tested larva stage ticks. The number 738 in Table 2 represents total number of tested ticks. We merged Table 1 and 2 and revised pie chart of Figure 1 according to the sample sizes.

Point 8: Line 84, the manuscript was written in a passive form, this sentence was active.

Response 8: We revised the sentence as active form (line 89-91).

Point 9: Line112-115, the distribution of Borrelia might also be a result of the small sample size.

Response 9: It is quite reasonable that the small number of the sample size may cause bias. But there is lots of previous reports on the distribution of Borrelia in ROK. We introduced nine references of them, and concluded as described.

Round 2

Reviewer 3 Report

The revised manuscript is much improved, and is appropriate to be published in Pathogen.
The title of Table 1 is not clear. I think the authors mean "Prevalence of Borrelia genospecies in engorged ticks by nested PCR for flaB".